# Ultra-Processed Foods and Metabolic Dysfunction-Associated Steatotic Liver Disease (MASLD): What Is the Evidence So Far?

**DOI:** 10.3390/nu17132098

**Published:** 2025-06-24

**Authors:** Eleni V. Geladari, Dimitris Kounatidis, Gerasimos Socrates Christodoulatos, Sotiria Psallida, Argyro Pavlou, Charalampia V. Geladari, Vassilios Sevastianos, Maria Dalamaga, Natalia G. Vallianou

**Affiliations:** 1Third Department of Internal Medicine, Evangelismos General Hospital, 10676 Athens, Greece; vsevastianos@gmail.com; 2Diabetes Center, First Propaedeutic Department of Internal Medicine, Medical School, National and Kapodistrian University of Athens, Laiko General Hospital, 11527 Athens, Greece; dimitriskounatidis82@outlook.com; 3Department of Microbiology, Sismanogleio General Hospital, 15126 Athens, Greece; gerchristod82@hotmail.com; 4Department of Microbiology, KAT General Hospital, 14561 Athens, Greece; psallidasotiria@gmail.com; 5First Department of Internal Medicine, Sismanogleio General Hospital, 15126 Athens, Greece; argirpavlou@gmail.com; 6Hellenic Society of Environmental and Climate Medicine, 17455 Athens, Greece; cgeladari@gmail.com; 7Department of Biological Chemistry, Medical School, National and Kapodistrian University of Athens, 11527 Athens, Greece; madalamaga@med.uoa.gr

**Keywords:** energy dense diet, metabolic dysfunction-associated steatotic liver disease, type 2 diabetes, ultra-processed foods

## Abstract

Ultra-processed foods (UPFs) are foods that have undergone extensive industrial processing, with the addition of emulsifiers and sweeteners together with various chemicals originating during preparation and the packaging procedures. UPFs are intended to be more palpable, long lasting, and easier to find and consume. However, their widespread use has been linked to various disorders, including insulin resistance, type 2 diabetes (T2D), obesity, and lately metabolic dysfunction-associated steatotic liver disease (MASLD). Given that MASLD is primarily driven by excessive fat accumulation in the liver and considering the high energy density and poor nutritional quality of UPFs, a plausible link has emerged between elevated UPF intake and increased MASLD risk. The aim of this review is to synthesize current data regarding the pathophysiological mechanisms underlying MASLD, the role of UPF overconsumption in its development, and potential strategies to prevent disease progression towards metabolic dysfunction-associated steatohepatitis (MASH), fibrosis, and cirrhosis. Special focus is placed on the contribution of UPFs to these processes, highlighting the importance of minimizing their consumption as supported by contemporary research.

## 1. Introduction

Ultra-processed foods (UPFs) comprise foods that have undergone extensive industrial processing, ranging from the addition of sweeteners and emulsifiers to the formation of chemicals during preparation and packaging. UPFs are intended to be more palatable, appealing to consumers, with longer lasting shelf times, and easier to consume [1,2]. According to the NOVA Classification System for Foods, there are four classes of foods: (I) unprocessed/minimally processed foods, which are foods as obtained from plants or animals, (II) unprocessed/minimally processed foods with the addition of salts, sugars, or oils for culinary purposes, (III) processed foods consisting of class I or class II foods that undergo further industrial manipulations to make them more tasty and better preserved, and (IV) UPFs, which consist of industrial formulations as already mentioned [1,2]. Figure 1 depicts several foods that are categorized according to the NOVA Classification Food System into four classes.

As the consumption of UPFs is constantly rising, there seems to be an association between UPFs and various metabolic disorders [3,4]. Among these, metabolic dysfunction-associated steatotic liver disease (MASLD) has been a prominent one. MASLD has been recently introduced as a definition that substitutes for non-alcoholic fatty liver disease (NAFLD). NAFLD had been widely used to describe liver fat accumulation that was not related to alcohol overconsumption. However, the amount of alcohol consumption differed between individuals and was not easily and accurately defined due to variability reasons. In addition, MASLD by definition is the previously described hepatic component of the metabolic syndrome. Thus, MASLD is not synonymous with NAFLD, as MASLD is defined as liver steatosis in the presence of at least one of the following five factors: (1) Body mass index (BMI) ≥ 25 Kg/m^2^ or waist circumference > 94 cm for men and >80 cm for women or adjusted for race; (2) fasting serum glucose ≥ 100 mg/dL or HbA1c ≥ 5.7% or 2 h post-load level of serum glucose ≥ 140 mg/dL on an oral glucose tolerance test (OGGT) or type 2 diabetes (T2D) or on treatment for T2D; (3) blood pressure ≥ 130/85 mmHg or on antihypertensive treatment; (4) serum triglycerides (TGs) ≥ 150 mg/dL or on hypolipidemic treatment; and (5) serum high-density lipoprotein (HDL) ≤ 40 mg/dL for men or ≤50 mg/dL for women or on hypolipidemic treatment [5,6]. This definition spares the plausible “stigma” of fatty liver disease in NAFLD and obviates the need for the existence of liver disease in the context of cardiometabolic factors [5,6]. It is noteworthy that 98% of the current registry cohort for NAFLD meet the criteria for MASLD [7].

Notably, Song et al. have pointed out that discrepancies between “the old NAFLD” definition and the newest MASLD one are minimal if considering findings from previous NAFLD studies under the MASLD definition invalid. Therefore, they concluded that we may use data from the NAFLD studies almost interchangeably with the MASLD trials that are yet to come at the forefront of future research [8]. Nowadays, MASLD has become an alarmingly increasing public health issue, worldwide. According to Le et al., MASLD cases accounted for 33.7% of the adult U.S. population in 2020 and will rise up to 41.4% of the adult U.S. population in 2050, according to the estimated trajectories [9]. As the MASLD menace is already present, it is mandatory to seek the factors associated with this public health problem.

The purpose of this review is to elaborate upon the potential link between overconsumption of UPFs and the development of MASLD. We aim to examine the current evidence for any causal relationship between the increased consumption of UPFs and the pathogenesis of MASLD. Apart from pathogenetic mechanisms, we will delve into the multi-faceted progression to MASLD, and we will thoroughly discuss the measures that should be taken in order to reduce the risk of developing MASLD.

## 2. Key Factors in MASLD Pathogenesis

MASLD is a multifactorial condition characterized by hepatic lipid accumulation, strongly linked to obesity, insulin resistance (IR), and T2D. In a cohort of 308 primary care patients with T2D, Balkhed et al. reported a MASLD prevalence of 59%, with 7% of individuals exhibiting advanced fibrosis and 1.9% cirrhosis [10]. Recently, Godoy-Matos et al. introduced the integrative concept of CARDIAL-MS (Cardio-Renal-Diabetes-Liver-Metabolic Syndrome), highlighting MASLD as a central component within a network of interconnected metabolic dysfunctions, particularly in individuals with T2D [11]. The pathogenesis of MASLD involves both genetic predisposition and environmental triggers. Among the latter, adherence to a Western dietary pattern, reduced physical activity, and hormonal imbalance have a substantial impact on disease development and progression [12,13].

Multiple studies have identified specific polymorphisms linked to increased risk of MASLD. In a Chinese cohort study, Pan et al. found significant correlations between apolipoprotein C3 (ApoC3) gene variants, specifically rs5128, rs2854116, and rs2854117, and increased MASLD susceptibility [14]. Furthermore, genome-wide association studies (GWASs) have implicated several single nucleotide polymorphisms (SNPs) in MASLD pathophysiology. Among these, variants in *PNPLA3* (patatin-like phospholipase domain-containing 3), *TM6SF2* (transmembrane 6 superfamily member 2), *MBOAT7* (membrane-bound O-acyltransferase domain-containing 7), and *GCKR* (glucokinase regulator) have been demonstrated to influence lipid metabolism and hepatic steatosis risk [15,16,17]. Interestingly, recent data support that both the MASLD-related genetic risk score (GRS) and the *PNPLA3* variant predict decade-long fibrosis progression, shaping distinct disease courses from midlife. Their combined use may also help identify patients at risk for hepatocellular carcinoma (HCC) in the absence of cirrhosis [18,19].

Diet is a key determinant in the development of MASLD, with UPFs comprising a core component of the Western dietary pattern. Globally, UPF consumption is rising due to UPF palatability, visual appeal, accessibility, convenience, and extended shelf life. The reduction in home cooking has further contributed to this trend [20]. UPFs are typically energy-dense yet nutrient-poor, characterized by elevated levels of added sugars, saturated fats, and sodium, and low in essential micronutrients and dietary fiber [20]. This dietary profile promotes weight gain and increases the risk of MASLD, as corroborated by several studies linking UPF intake to metabolic disturbances [21,22]. Obesity, a principal risk factor for MASLD, is closely associated with the high caloric density of UPFs [23,24].

The hypothesis that excessive consumption of UPFs contributes to the pathogenesis of MASLD has likely emerged from the observed parallel rise in UPF consumption and MASLD prevalence. According to a recent report by the Global Food Research Program, UPFs account for over half of total calorie intake in the United States, the United Kingdom, and Canada, and for up to 40% in other high- and middle-income countries [25]. Moreover, Taylor et al. have advanced the Personal Fat Threshold (PFT) hypothesis, which posits that once an individual’s subcutaneous fat storage capacity is exceeded, ectopic lipid accumulation ensues. This model explains the occurrence of MASLD in individuals with a normal BMI [26].

Beyond their contribution to excess caloric intake and weight gain, UPFs also contribute to systemic inflammation and oxidative stress, mechanisms integral to MASLD progression [27,28]. Growing evidence links high UPF consumption to elevated levels of inflammatory biomarkers, including interleukin-6 (IL-6), tumor necrosis factor-alpha (TNF-α), high-sensitivity C-reactive protein (hs-CRP), and increased white blood cell count (WBC) [27,28]. Data also indicate that higher consumption of UPFs is associated with reduced levels of physical activity [29]. Since physical inactivity can contribute to weight gain and the development of MASLD, a sedentary lifestyle driven by excessive UPF intake may perpetuate a vicious cycle that exacerbates MASLD progression [30].

In recent years, research has increasingly highlighted the detrimental effects of endocrine-disrupting chemicals (EDCs) on human health. EDCs comprise a diverse group of compounds known to interfere with hormonal systems and contribute to adverse health outcomes [31,32]. Notably, dietary patterns characterized by high consumption of UPFs have been associated with increased exposure to EDCs. The presence of EDCs in UPFs is largely attributed to the use of additives and contamination from various compounds during processing and packaging [33,34].

EDCs are widely recognized for their ability to interfere with hormonal receptors, such as estrogen receptors (ERs) and androgen receptors (ARs). Among EDCs, phthalates and bisphenols, particularly bisphenol A (BPA), are the most extensively studied. Phthalates, esters of phthalic acid used as plasticizers, commonly migrate into UPFs during packaging, especially when exposed to high temperatures or prolonged storage periods [4,35]. According to Yang et al., phthalates have recently been associated with the development of MASLD in individuals with IR, prediabetes, or T2D [36]. BPA, another commonly used plastic, can leach into food during heating or under extreme pH conditions. It has been linked to disruptions in pancreatic alpha and beta cell function, contributing to the development of IR and T2D [4,37]. Interestingly, recent research has shed light on the role of a subset of EDCs, known as metabolism-disrupting chemicals (MDCs), which may exert obesogenic effects by impairing mitochondrial function in adipocytes. This dysfunction promotes inflammation and oxidative stress, ultimately activating adipose tissue and promoting the release of adipokines and pro-inflammatory cytokines, which are key drivers in the development of MASLD, obesity, and T2D [37,38]. The term “oxinflammation” has been proposed to describe this pathological state, characterized by the synergistic interaction between oxidative stress and chronic low-grade inflammation, shaping a shared mechanistic link among these conditions [38]. Figure 2 presents key contributors in MASLD development.

## 3. Pathophysiological Mechanisms Linking Ultra-Processed Foods to MASLD

As previously discussed, UPFs are typically rich in energy due to their high content of added sugars, fats, and salt, while being low in essential nutrients such as vitamins, minerals, and dietary fibers. This nutritional profile favors weight gain and increased adiposity [39]. Excess adipose tissue leads to elevated levels of circulating free fatty acids (FFAs), which are transported to the liver and contribute to inflammation, oxidative stress, and endoplasmic reticulum stress (ERS), all of which exacerbate MASLD and promote its progression to metabolic dysfunction-associated steatohepatitis (MASH) [23,24]. A hallmark of MASLD is enhanced de novo lipogenesis (DNL) in hepatocytes, accompanied by increased lipolysis in white adipose tissue (WAT) [23,24]. DNL is upregulated in MASLD and mediated by key lipogenic enzymes such as acetyl-CoA carboxylase (ACC) and fatty acid synthase (FAS). The activity of these enzymes is stimulated by insulin and ghrelin, while adiponectin exerts inhibitory effects [40]. The combined result of FFA influx and upregulated DNL leads to intracellular accumulation of lipid droplets and the formation of lipotoxic intermediates in hepatocytes.

When lipid overload exceeds the protein-folding capacity of the endoplasmic reticulum, unfolded proteins accumulate, triggering ERS and activation of the unfolded protein response (UPR). Chronic UPR activation results in inflammation and oxidative stress via the generation of reactive oxygen species (ROS), and eventually hepatocyte apoptosis [41,42]. ROS further activate the NLR family pyrin domain containing 3 (NLRP3) inflammasome, which processes pro-caspase-1 into active caspase-1, thereby promoting the production of pro-inflammatory cytokines interleukin-1β (IL-1β) and interleukin-18 (IL-18), key mediators in the inflammatory cascade of MASH [41,42]. Furthermore, CCN3 (nephroblastoma overexpressed, NOV), an adipokine secreted by adipose tissue and a member of the CCN protein family, has been implicated in hepatic inflammation and fibrosis. Afrisham et al. demonstrated increased serum CCN3 levels in MASLD patients, which correlated with elevated interleukin-6 (IL-6) and tumor necrosis factor-alpha (TNF-α), compared to healthy controls [43,44].

In addition to EDCs, UPFs are also rich in advanced glycation end-products (AGEs). AGEs are typically generated during industrial processing, especially under conditions of excessive heating, and have recently been implicated in the pathogenesis of MASLD [45,46]. AGEs interact with their receptor RAGE in hepatocytes, initiating a signaling cascade that leads to increased mitochondrial ROS production and hepatocyte ballooning [47,48,49]. RAGE activation also triggers transcription factors such as nuclear factor kappa B (NF-κB), facilitating the expression of genes related to inflammation, including NLRP3, pro-IL-1β, and pro-IL-18, thereby enhancing inflammasome activation [49,50,51,52]. A population-based study by Jahromi et al. linked higher dietary AGEs to increased MASLD prevalence among Iranian adults [53].

Beyond metabolic and inflammatory pathways, UPFs have been shown to alter the composition of the gut microbiota. The term “gut microbiota” refers to the diverse community of bacteria, viruses, fungi, and archaea residing predominantly in the human gastrointestinal tract. Under normal conditions, a state of homeostasis exists between these microbial populations and the host. On the contrary, UPFs may facilitate reductions in microbial diversity, as they may lead to lower levels of beneficial species such as *Akkermansia muciniphila* and *Faecalibacterium prausnitzii* [17,54,55]. Disruption of this balance, also known as gut dysbiosis, is characterized by compositional and functional changes in the gut microbiota [17].

The gut–liver axis plays a central role in MASLD pathophysiology. In MASLD, the intestinal epithelial barrier is compromised, exhibiting reduced expression of tight junction proteins. This dysfunction, exacerbated by emulsifiers and additives found in UPFs, leads to increased intestinal permeability—a phenomenon termed “leaky gut” [17,54,55,56]. The development of “leaky gut” may also be influenced by the elevated levels of EDCs and AGEs found in UPFs [35,47,48,49]. As a result, lipopolysaccharides (LPSs) from Gram-negative bacteria translocate into the bloodstream and activate Toll-like receptors (TLRs), particularly TLR4 and TLR9, on hepatic and immune cells. This activation promotes inflammatory responses that drive the progression to MASH [17,54,55,56]. Additionally, the gut microbiota metabolite trimethylamine (TMA), derived from dietary components such as choline, is absorbed and converted in the liver to trimethylamine-N-oxide (TMAO) by hepatic flavin monooxygenases. TMAO has been implicated not only in MASLD development, but also in obesity, T2D, and cardiovascular diseases [17,57].

Figure 3 depicts the key pathogenetic pathways behind the development of MASLD/MASH due to the overconsumption of UPFs. Subsequently, Figure 4 illustrates the main pathophysiological mechanisms through which UPFs may contribute to the development or exacerbation of MASLD/MASH, highlighting the role of the gut–liver axis.

## 4. Associating Ultra-Processed Foods with MASLD

### 4.1. Current Concepts

Zhang et al. have shown an association between overconsumption of UPFs and an increased risk of severe NAFLD [58]. Indeed, in their prospective cohort study among 143,073 individuals from the UK Biobank, they used the NOVA Classification for Food System and 24 h dietary recall data. The severity of NAFLD was determined as NAFLD using the International Classification of Diseases, Tenth Revision (ICD-10) definition necessitating hospitalization or even death from NAFLD. Within a median follow-up of 10.5 years, 1945 participants developed severe NAFLD. Notably, Zhang et al. reported that for a 10% increase in the consumption of UPFs, there was a significant increase in the incidence of severe NAFLD. In their quartile analysis, they found that participants in the first quartile, who had the highest consumption of UPFs, exhibited a 1.38 to 1.49 higher risk of developing severe NAFLD when compared to participants in the fourth quartile, who had the lowest UPF consumption. However, their study, even though illustrating an association between higher consumption of UPFs and an increased risk of severe NAFLD, cannot justify causality, as it was a prospective cohort study and not a randomized controlled trial (RCT). Additionally, they used 24 h dietary recall data and not a Food Frequency Questionnaire (FFP) [58]. Furthermore, the definition of NAFLD is slightly different from MASLD. Nevertheless, it is noteworthy that both NAFLD and MASLD are usually assessed by different biomarkers/scores and imaging modalities on a global scale, whereas the gold standard for defining NAFLD/MASLD, i.e., liver biopsy, is not routinely performed. Notwithstanding these parameters, Zhang et al. have demonstrated a relationship between higher consumption of UPFs and an increased risk of severe NAFLD [58].

Zhao et al. in their prospective cohort study involving 173,889 individuals also from the UK Biobank, using NOVA, 24 h dietary recall data, and the ICD-10 definition for NAFLD, have linked higher consumption of UPFs to increased risk of NAFLD. In particular, after a median of 8.9 years of follow-up, they reported 1108 cases of NAFLD, 350 cases of liver fibrosis/cirrhosis, and 550 cases of severe liver diseases in general. They also demonstrated that individuals in the highest quartile of UPF consumption had an increased risk for developing NAFLD (Hazards Ratio HR Quartile 4 vs. Quartile 1: 1.43; 95% CI: 1.21, 1.70; *p* < 0.001) [59]. Previously, in 2022, another prospective study, having enrolled 16,168 adults from the Tianjin Chronic Low-grade Systemic Inflammation and Health (TCLSIH) study in China using an FFQ, had also documented a correlation between higher consumption of UPFs and NAFLD [60]. In addition, Zhao et al., in a national survey using data from the National Health and Nutrition Examination Survey (NHANES) during 2017–2018, examined whether higher consumption of UPFs was associated with an increased risk of NAFLD [61]. They included 806 adolescents and 2734 adults from this period and used two sets of 24 h dietary recall data. They also defined NAFLD by using the vibration-controlled transient elastography (VCTE) method. A cut-off in VCTE controlled attenuation parameter (CAP) of 285 dB/m was the CAP above which the patient was defined as having NAFLD. Furthermore, clinically significant fibrosis (CSF) was considered ≥ 8.6 kPa. They concluded that in both adolescents and adults, there was a significant relationship between higher consumption of UPFs and an increased risk of NAFLD [61].

Apart from these cohort studies, there are some systematic reviews and meta-analyses on the association between overconsumption of UPFs and the risk for developing NAFLD, mainly during the past five years. Amongst them, Henney et al., in their meta-analysis that included nine studies, found a strong correlation between higher consumption of UPFs and the risk of NAFLD. They pointed out that they observed a dose–response association between consumption of UPFs and the development of NAFLD [62]. Further, Grinshpan et al. in their systematic review have commented on the relationship between higher consumption of UPFs and risk of NAFLD. They have included 15 studies, which apart from an association with obesity and T2DM, have shown a relationship between higher consumption of UPFs and NAFLD in three out of six studies [63]. Table 1 depicts major studies regarding UPFs and the risk of MASLD.

### 4.2. Future Perspectives

Although there is enough evidence correlating overconsumption of UPFs with increased risk of MASLD, RCTs, which may confirm causality, are lacking. Moreover, further large-scale studies are needed to explore the current findings, after adjusting for potential confounding factors. Apart from abdominal obesity, there are other mechanisms involved in the pathogenesis of MASLD, which are yet to be thoroughly studied and elaboratively discussed. For the time being, most researchers recommend that the overconsumption of UPFs should be decreased in order to reduce the risk of MASLD [64,65,66,67,68,69,70,71,72,73,74]. This holds true especially for patients with abdominal obesity and T2D. However, increased consumption of UPFs should be avoided not only for mitigating the risk of MASLD, but also as a nutritional approach that patients with MASLD have to pursue in order to regress the progression to MASH and liver fibrosis/cirrhosis [75,76,77]. Notably, the definition of “overconsumption of UPFs” may be difficult to assess unless a quantitative FFQ is thoroughly used in future studies. Moreover, 139,443 chemical compounds have been estimated to exist in foods, with this number constantly rising; therefore, the complexity regarding food research and relative estimations is apparent [78]. Nevertheless, UPFs are widely available; thus, we have to point out that there is an increased possibility that molecules responsible for the development of MASLD could aggregate in the human body. These compounds derived from different UPFs may result in accumulation and a subsequent augmentation of their detrimental effects on human health. Children, adolescents, and the elderly are particularly vulnerable in this context.

It is noteworthy that the Mediterranean diet has been linked to halting progression of MASLD [79,80,81]. According to Garcia et al., individuals who follow the Mediterranean diet have been demonstrated to exhibit lower levels of UPF consumption. This reduction of UPF consumption results in an enhancement in anti-inflammatory and antioxidant intake [21]. Therefore, Garcia et al. have concluded that following a Mediterranean diet, which is rich in whole grains, fruits, vegetables, legumes, nuts, and fish, may lead to a reduction in the intake of UPFs together with an increase in the total intake of foods with antioxidant properties [21]. Interestingly, red meat and red meat products, which are processed or UPFs, are known for their high content in saturated fatty acids (SFAs), which when in excess in the human body may lead to ectopic fat accumulation, especially in the liver [82]. Thus, red meat, and in particular processed red meat, has been associated with the development of MASLD [82,83,84]. Processed red meat together with sugary beverages and processed sweets all contribute to the increased risk of MASLD [85,86,87,88]. In sharp contrast, the Mediterranean diet, which is poor in the aforementioned foods, has been suggested to mitigate the progression of MASLD to MASH and liver fibrosis/cirrhosis [89,90,91,92,93,94]. Apart from diet, other modifiable risk factors for MASLD, such as physical inactivity, should be discouraged. As MASLD, obesity, and lack of exercise are interconnected, it is highly advisable to incorporate exercise as an integral part of people’s daily routine. In addition, as alcohol may augment the risk of liver damage among patients with MASLD, its abstinence is of paramount importance [95]. Table 2 summarizes suggestions for improving health status among patients with MASLD [3].

## 5. Conclusions

In conclusion, there is growing evidence associating overconsumption of UPFs with a higher risk of MASLD. A relationship between increased consumption of UPFs and the development of MASLD seems to be likely on the grounds of pathophysiology as well as current available data. Therefore, it would be prudent to decrease the consumption of UPFs, especially among patients with MASLD, in order to mitigate progression to MASH and liver fibrosis/cirrhosis. Nevertheless, large-scale studies, especially RCTs, in this field are eagerly anticipated to further confirm or not the aforementioned recommendations.

## Figures and Tables

**Figure 1 nutrients-17-02098-f001:**
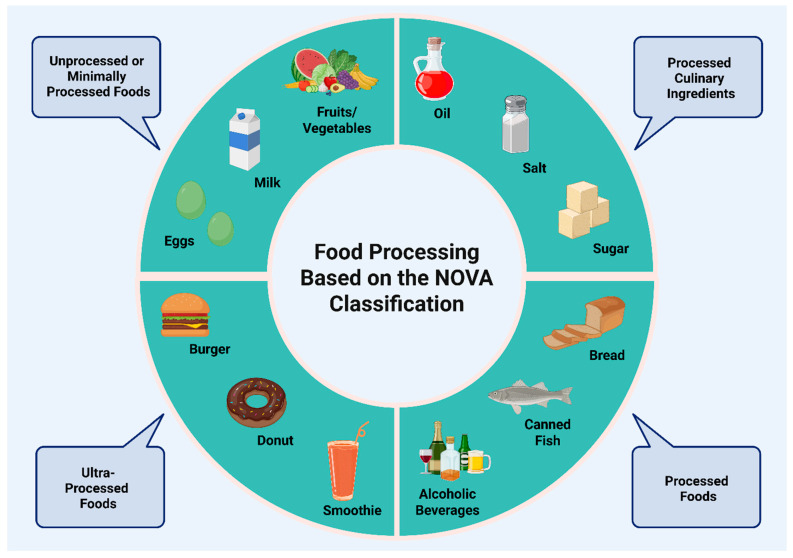
Examples of food types according to degree of processing based on NOVA classification. Created in BioRender. Kounatidis, D. (2025) https://BioRender.com/xvxzpme. Assessed on 20 June 2025.

**Figure 2 nutrients-17-02098-f002:**
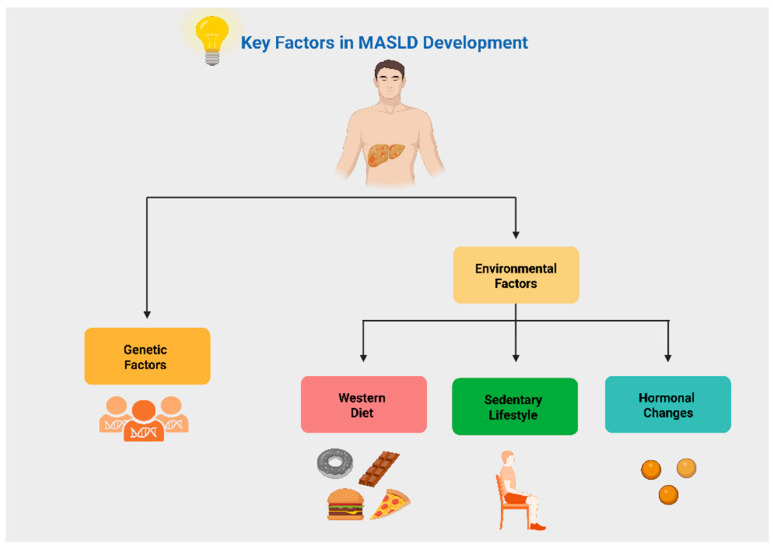
Key factors in the development of metabolic dysfunction-associated steatotic liver disease (MASLD). Created in BioRender. Kounatidis, D. (2025) https://BioRender.com/dryfly9. Assessed on 20 June 2025.

**Figure 3 nutrients-17-02098-f003:**
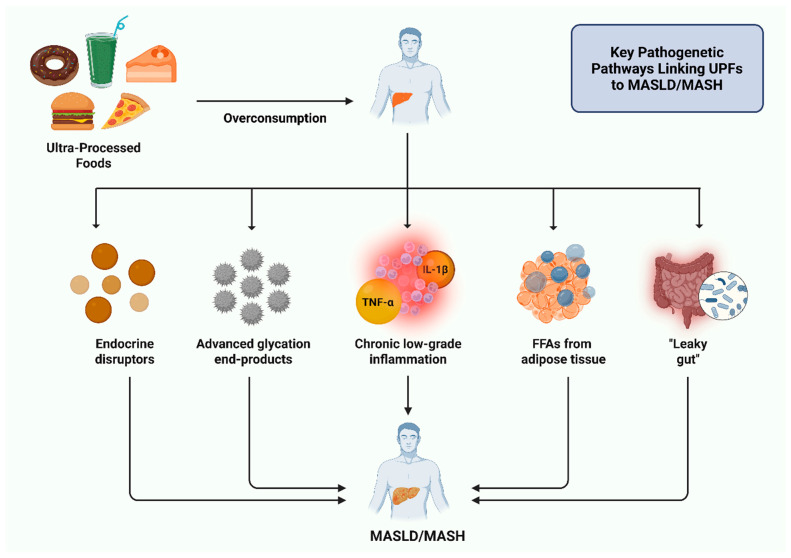
Central pathophysiological pathways through which excessive consumption of ultra-processed foods contributes to the development and progression of metabolic dysfunction-associated steatotic liver disease/steatohepatitis (MASLD/MASH). Abbreviations: FFAs: free fatty acids; IL-1β: interleukin-1β; MASH: metabolic dysfunction-associated steatohepatitis; MASLD: metabolic dysfunction-associated steatotic liver disease; TNF-α: tumor necrosis factor-alpha; UPFs: ultra-processed foods. Created in BioRender. Kounatidis, D. (2025) https://BioRender.com/qfzqa4q. Assessed on 22 June 2025.

**Figure 4 nutrients-17-02098-f004:**
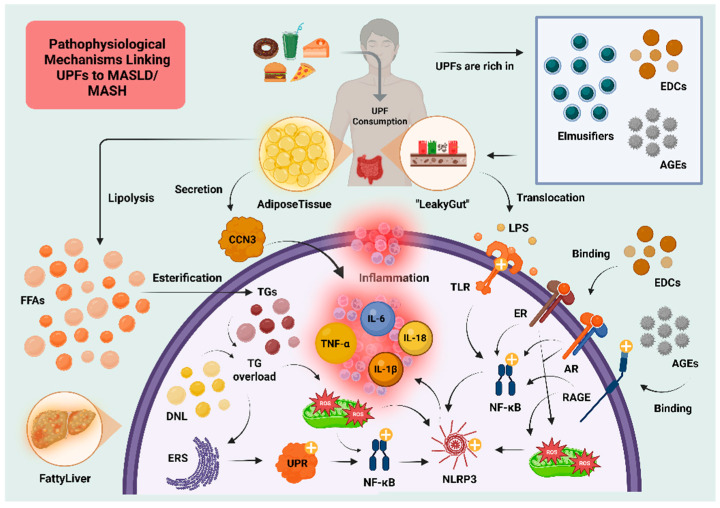
Main pathophysiological mechanisms linking ultra-processed food (UPF) consumption with the development and progression of metabolic dysfunction-associated steatotic liver disease/steatohepatitis (MASLD/MASH). Excessive intake of UPFs contributes to body weight gain and adipose tissue expansion, leading to increased lipolysis and elevated circulating FFAs. In the liver, FFAs are esterified into TGs, contributing to hepatic steatosis. Simultaneously, DNL is upregulated in the context of MASLD. Hepatocellular TG overload induces ERS and mitochondrial oxidative stress, both of which trigger inflammatory pathways via activation of the UPR and ROS production, respectively. These stress responses converge to activate the NLRP3 inflammasome, with NF-κB playing a central regulatory role. Additional pro-inflammatory signals arise from EDCs and AGEs found in UPFs, as well as from the novel adipokine CCN3 secreted by dysfunctional adipose tissue. Furthermore, UPF-derived emulsifiers, along with EDCs and AGEs, disrupt gut barrier integrity, promoting the development of a “leaky gut”. This facilitates the translocation of LPS into the systemic circulation and subsequently to the liver, where TLRs mediate additional inflammatory responses. The cumulative impact of these pathways highlights the gut–liver axis as a key contributor to MASLD/MASH pathogenesis. Abbreviations: AGEs: advanced glycation end-products; AR: androgen receptor; CCN3: nephroblastoma overexpressed, NOV; DNL: de novo lipogenesis; EDC: endocrine-disrupting chemical; ER: estrogen receptor; ERS: endoplasmic reticulum stress; FFA: free fatty acid; IL: interleukin; LPS: lipopolysaccharide; MASLD: metabolic dysfunction-associated steatotic liver disease; MASH: metabolic dysfunction-associated steatohepatitis; NF-κB: nuclear factor kappa-light-chain-enhancer of activated B cells; NLRP3: NLR family pyrin domain containing 3; RAGE: receptor for advanced glycation end-products; ROS: reactive oxygen species; TG: triglyceride; TLR: toll-like receptor; TNF-α: tumor necrosis factor-alpha; UPF: ultra-processed food; UPR: unfolded protein response; +: activation. Created in BioRender. Kounatidis, D. (2025) https://BioRender.com/blmuucp. Assessed on 22 June 2025.

**Table 1 nutrients-17-02098-t001:** Major studies examining the association between ultra-processed food consumption and the risk of MASLD.

Author/Year	Population	Study	Findings	Remarks
Zelber-Sagi et al., 2018 [64]	789 individuals, aged 40 to 70 y.o. with a valid FFQ, and 357 with a valid meat questionnaire, in Israel between 2013 and 2015	Cross-sectional Study	Higher intake of processed red meat was associated with increased odds for NAFLD as well as IR For red meat OR:1.47 95% CI: 1.04–2.09, *p* = 0.031. For processed red meat OR: 1.55, 95% CI: 1.07–2.23, *p* = 0.020.	Increased consumption of red processed meat was related to higher odds for NAFLD. The authors conclude that avoiding red processed meat and unhealthy cooking methods may alleviate NAFLD.
Noureddin et al., 2020 [65]	2974 patients with NAFLD, with and without cirrhosis, with a valid FFQ, aged 45 to 75 y.o., in Hawai and California, U.S.A., between 1993 and 1996	Nested case-control analysis within the MEC (Multiethnic cohort) prospective study with >215,000 participants	Higher intake of red meat and processed red meat was associated with NAFLD (*p* = 0.010 and *p* = 0.004, respectively).	The association between higher red meat and processed red meat intake and increased NAFLD was more prominent, especially among patients with NAFLD and cirrhosis. The authors concluded that by reducing the intake of red meat and processed red meat, patients could decrease the risk of NAFLD, and in particular NAFLD and advanced liver disease.
Yari et al., 2020 [66]	143 adult patients with newly diagnosed NAFLD and 471 controls with a valid FFQ in Iran	Case-control study	Sweet energy-dense nutrient-poor snacks were moderately associated with a higher risk of NAFLD	The authors conclude that there might be a moderate association between energy-dense snacks and risk of NAFLD.
Ivancovsky-Wajcman et al., 2021 [67]	789 participants, among whom 305 individuals were diagnosed with NAFLD, with a valid FFQ in Israel	Cross-sectional study	Higher consumption of UPFs was associated with increased odds for METS (OR: 1.88 95% CI: 1.31–2.71, *p* = 0.001).	Increased consumption of UPFs is related to a higher risk of METS, and among patients with NAFLD, it is associated with an increased risk of NASH.
Odegaard et al., 2022 [68]	5115 participants, aged 18 to 30 y.o., between 1985 and 1986 were enrolled from 4 cities in the USA	Prospective multicenter cohort study (the CARDIA Study)	Participants were followed for 25 years, and there was a significant association between MAFLD in middle age among individuals with more frequent consumption of fast food.	Although no FFQ had been used, participants answered a question regarding frequent or infrequent fast food consumption, six times during the 25 years of the follow-up. Increased VAT, liver fat, and odds of MAFLD were related to a more frequent consumption of fast food during the 25 years.
Rahimi-Sakak et al., 2022 [60]	196 patients with NAFLD and 803 controls, aged >18 y.o., with a valid FFQ, in Iran	Case-control study	Patients with the highest quartile of processed red meat consumption had a 3.28 times higher risk of NAFLD, when compared to the lowest quartile of processed red meat consumption (OR: 3.28, 95% CI: 1.97–5.46, *p* < 0.001).	The authors conclude that there seems to be a relationship between higher consumption of processed red meat and the risk of NAFLD, which requires further evaluation.
Friden et al., 2022 [69]	285 individuals, aged 50 y.o., with a valid FFQ in 2010, in Uppsala, Sweden	Cross-sectional study based on data taken from a prospective population-based cohort	There was an association between consumption of UPFs that were rich in energy and carbohydrates with VAT. This association was more significant among women than men.	Although energy intake from UPFs was not associated with liver fat nor SAT, there was an association between UPFs and VAT, especially among women rather than men.
Zhang et al., 2022 [52]	16,168 participants, with a valid FFQ, aged 18 to 90 y.o., in China	Prospective study	Among the studied population, i.e., 16,168 individuals, there were 3752 cases of NAFLD during 56,935 person-years of follow-up.	The study documented a relationship between higher UPF consumption and increased risk of NAFLD. The authors conclude that UPFs may be a modifiable factor regarding risk of NAFLD.
Konieczna et al., 2022 [22]	5867 patients with METS, with a valid FFQ, aged 55 to 75 y.o. were followed for 1-year enrollment during 2013 and 2016 in Spain	Prospective analysis from the PREDIMED Plus trial	A 10-fold increment in the daily consumption of UPFs was associated with a significantly higher FLI as well as HSI.	The authors conclude that a higher consumption of UPFs was associated with increased levels of NAFLD biomarkers among patients with overweight/obesity and METS.
Zhao et al., 2023 [61]	806 adolescents and 2734 adults were included between 2017 and 2018 from the NHANES study, with 2 days of 24 h recall data, in the U.S.A.	Cohort study	Higher consumption of UPFs was associated with increased odds of NAFLD (In adolescents OR: 2.34 95% CI: 1.01–5.41, *p* = 0.15, in adults OR: 1.72 95% CI: 1.01–2.93, *p* = 0.002)	A higher consumption of UPFs was associated with increased risk of NAFLD, which was 68% and 71% mediated in adolescents and adults by increased body fat.
Zhang et al., 2024 [58]	143,073 participants from the U.K. Biobank, with 24 h recall data in the U.K.	Prospective cohort study	After a median of 10.5 years of follow-up, 1445 participants developed NAFLD in its severe form, as hospitalization or death due to NAFLD or NASH.	The authors concluded that by decreasing the consumption of UPFs there may be a reduction in NAFLD.
Zhao et al., 2024 [59]	173,889 participants from the U.K. Biobank, aged 40 to 69 y.o., with 24 h dietary recall data in the U.K.	Prospective cohort study	After a median of 8.9 years of follow-up, the authors documented an association between higher consumption of UPFs and an increased risk of NAFLD, as well as liver fibrosis/cirrhosis.	The authors commented that by reducing consumption of UPFs, there may be an amelioration in NAFLD and liver fibrosis/cirrhosis and an improvement in liver health status.

Abbreviations: 95% CI: 95% confidence interval; FFQ: food frequency questionnaire; FLI: fatty liver index; HSI: hepatic steatosis index; MAFLD: metabolic associated fatty liver disease; MEC: multiethnic cohort; METS: metabolic syndrome; OR: odds ratio; SAT: subcutaneous adipose tissue; U.K.: United Kingdom; VAT: visceral adipose tissue; y.o.: years old.

**Table 2 nutrients-17-02098-t002:** Review of recommendations regarding UPFs and the prevention and avoidance of progression of MASLD to MASH and liver fibrosis/cirrhosis.

Encourage drinking fresh and safe water instead of beverage consumption
Substituting red meat and red meat products with fish or poultry
Encouraging learning programs about what UPFs are and how to distinguish them in order for parents, children/adolescents, and the elderly to avoid them
Encouraging the consumption of more fruits, vegetables, legumes, nuts, and whole grains
Encouraging fresh cooking and cooking skills at home
Educational programs regarding the potential harms of UPFs for parents, at school, and possibly at work
Legislations for detailed labelling of ingredients/calories of UPFs
Protecting parents and especially children/adolescents against uncontrolled advertising of UPFs
Legislations for extra taxes on UPFs

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
