# Peer review of "Ultra-Processed Foods and Metabolic Dysfunction-Associated Steatotic Liver Disease (MASLD): What Is the Evidence So Far?"

_nutrients, 2025, doi:10.3390/nu17132098_

Round 1

Reviewer 1 Report

Comments and Suggestions for Authors

This literature review provides significant insights into the impact of ultra-processed foods on liver disease. The following suggestions are proposed:

Line 68 

To improve clarity, add citation [6] at the end of the sentence.

Line 98

If possible, a diagram could be included to visually illustrate the pathophysiological processes involved in MASLD, particularly the impact of a Western diet on hepatic lipid accumulation, gut function, and the gut–liver axis, as described in the text.

Line 114

To improve clarity, may be can consider expanding on how DNL leads to MASLD, particularly the roles of enzymes like ACC and FAS.

Line 284

Table 1 For improve readability, can consider formatting the table in landscape orientation and organizing the information into the following columns: Author/Year, Study Design, Population, Findings, and Remarks.

Reviewer 2 Report

Comments and Suggestions for Authors

The authors have presented a review on

 Metabolic Dysfunction-Associated Steatotic Liver Disease (MASLD) due to exposure to ultra processed food. The authors have reviewed very well the association of selected  Endocrine disruptors with MASLD/NAFLD. The NAFLD/MASLD is a disease at the interface of metabolic syndrome and diabetes. It would be wise to point out the role of EDC with Diabetes and diabetes mediated MASLD/NAFLD. Appropriately, authors should insert one to two paragraph the association of EDC chemicals with Diabetes and perform review of diabetes association with MASLD/NAFLD

Pathogenesis of MASLD Section:

The authors covered a number of genes involved in MASLD but ignored  the role of CCN3/NOV (Cellular communication network factor 3). It is recommended that authors insert at least one to two sentences on the novel adipokine.

Saghir SA, Shams N, Veliz L, Alfuraih S, Omidi Y, Barar J, et al. Non-alcoholic fatty liver disease: Genetic susceptibility. Arch Clin Toxicol. 2024;6(1):33-47.

Line 204-206: Does the binding of AGEs with RAGE on immune cells cause immune activation and release of inflammatory cytokines? Authors are advised to describe further with 3-4 lines. “More specifically, increased amounts of AGEs may lead to activation of the hepatic receptor of AGEs (RAGE), a fact which is related to a cascade of production of ROS and ultimately, to hepatic cell ballooning [40].”

Reviewer 3 Report

Comments and Suggestions for Authors

The text reviews how eating too much ultra-processed food is connected with higher risk of getting metabolic dysfunction-associated steatotic liver disease. It explains that these foods are very processed with added chemicals, sweeteners and fats, and they are high in energy but low in good nutrients. The paper shows that this type of food makes people gain weight and changes how fat is stored in the liver. This can lead to liver inflammation and more serious liver problems. The review gives many studies that found people who eat more ultra-processed food have more risk of liver disease. It says the Western diet and easy access to these foods make this a growing health problem worldwide. The paper also explains how chemicals from food packaging and processing can harm hormones and make the problem worse. It suggests that eating less ultra-processed food and following diets like the Mediterranean diet can help protect the liver. The review says more big studies are needed but now it is smart to eat more fresh and less processed food to avoid liver disease.
The text can be improved in many ways to make it stronger and more practical. First, it should add more results from good randomized controlled trials, because now most evidence is from cohort or observational studies which only show connection, not clear proof. Second, it should show how big is the risk for different groups, for example children, older people or people with other diseases. Third, it should give more simple and clear advice for everyday life, not just say “eat less ultra-processed food” but also suggest what to eat instead, how to cook fast meals at home, or how to read food labels. Fourth, the paper can explain more how governments and schools can help limit ultra-processed food, like taxes on bad food, better food rules in schools and ads. Fifth, some parts have hard scientific words and long sentences, so shorter sentences and simple words would help non-expert readers understand. Sixth, it could add a short summary or checklist at the end with main practical tips. Seventh, more figures, infographics or charts would help readers see the main ideas quickly. Eighth, it could compare the effect of ultra-processed food with other risk factors for liver disease, like alcohol or lack of exercise. Ninth, more clear information on how much ultra-processed food is “too much” would help people know when they are at risk. Last, it could include ideas for future research and what questions are still open. This would show what is missing and what scientists need to study next.
